# Efficient retrosynthetic planning with MCTS exploration enhanced A* search
Dengwei Zhao [1], Shikui Tu [1] ✉ & Lei Xu [1,2] ✉

Retrosynthetic planning, which aims to identify synthetic pathways for target molecules from starting materials, is a fundamental problem in synthetic chemistry. Computer-aided retrosynthesis has made significant progress, in which heuristic search algorithms, including Monte Carlo Tree Search (MCTS) and A* search, have played a crucial role. However, unreliable guiding heuristics often cause search failure due to insufficient exploration. Conversely, excessive exploration also prevents the search from reaching the optimal solution. In this paper, MCTS exploration enhanced A* (MEEA*) search is proposed to incorporate the exploratory behavior of MCTS into A* by providing a look-ahead search. Path consistency is adopted as a regularization to improve the generalization performance of heuristics. Extensive experimental results on 10 molecule datasets demonstrate the effectiveness of MEEA*. Especially, on the widely used United States Patent and Trademark Office (USPTO) benchmark, MEEA* achieves a 100.0% success rate. Moreover, for natural products, MEEA* successfully identifies bio-retrosynthetic pathways for 97.68% test compounds.

For a given target molecule, retrosynthetic planning aims to identify a feasible and cost-effective synthetic route from the enormous chemical reaction space, using known or commercially available building block molecules[1]. It is one of the fundamental problems in synthetic chemistry and plays a critical role in a wide range of applications, such as drug design[2,3] and material science[4]. The expenditure required to bring a single drug to market exceeds 2 to 3 billion dollars, which is primarily allocated toward the preceding discovery phase and clinical trials[5]. Using the data-driven retrosynthetic tools during the discovery phase to accelerate and reduce failures in the synthesis of new drug molecules has attracted significant interest in recent years[6].

Considering that the synthesis of most molecules typically requires multiple chemical reactions and the number of possible chemical transformations is vast, retrosynthesis planning is a challenging task even for experienced chemists. The difficulty of synthesizing a molecule is highly influenced by its structure complexity and available building blocks. In some cases, the identification of a viable synthetic pathway requires weeks of intense effort by human experts. Computer-assisted synthesis planning could assist chemists in expediting the identification of high-quality synthetic pathways[7], which commonly employs a search algorithm guided by a single-step retrosynthetic model. With the development of artificial intelligence technology, machine learning-based algorithms have demonstrated remarkable performance, learning to generate recommendations automatically. These algorithms usually perform the single-step retrosynthesis as a classification task based on reaction templates[8–11]. Another alternative approach is to leverage

sequence-to-sequence models in natural language processing, treating the one-step retrosynthetic task as a translation problem between products and reactants[12–17]. Recently, semi-template-based methods are proposed to predict the reaction center dictating a reaction firstly via graph neural networks, and then translate the resulting intermediate synthons into reactants via translation models[18–20]. Single-step models predict the most promising chemical reactions that can directly synthesize the target molecule. Although a greedy strategy or other simple exploration strategies can be employed to generate a complete synthetic route[21–23], search algorithms are usually employed to extent the single-step model to full route design, providing higher-quality solutions with improved efficiency. Due to the large number of possible precursors at each step, the single-step model serves as a guiding search strategy to prevent combinatorial explosion. Figure 1 illustrates how the traditional chemical synthetic pathway is transformed into the search tree. The nodes in the search tree represent the synthetic position, containing all molecules required to synthesize the target molecule in the root state. The target molecule can be synthesized if all molecules within a state are available building blocks. The edges in the search tree correspond to chemical reactions that induce state transitions between the two connected nodes.

An excellent searching strategy can reveal meaningful retrosynthesis efficiently from the large chemical space. Traditional heuristic methods, such as depth-first, have poor performance and the quality of their suggested pathway is not guaranteed. Inspired by the excellent performance of deep reinforcement learning in mastering combinatorial games[24], Segler et al.[25]

[1]Department of Computer Science and Engineering, Shanghai Jiao Tong University, Shanghai, China. [2]Guangdong Institute of Intelligence Science and Technology, Zhuhai, China. ✉e-mail: tushikui@sjtu.edu.cn; leixu@sjtu.edu.cn

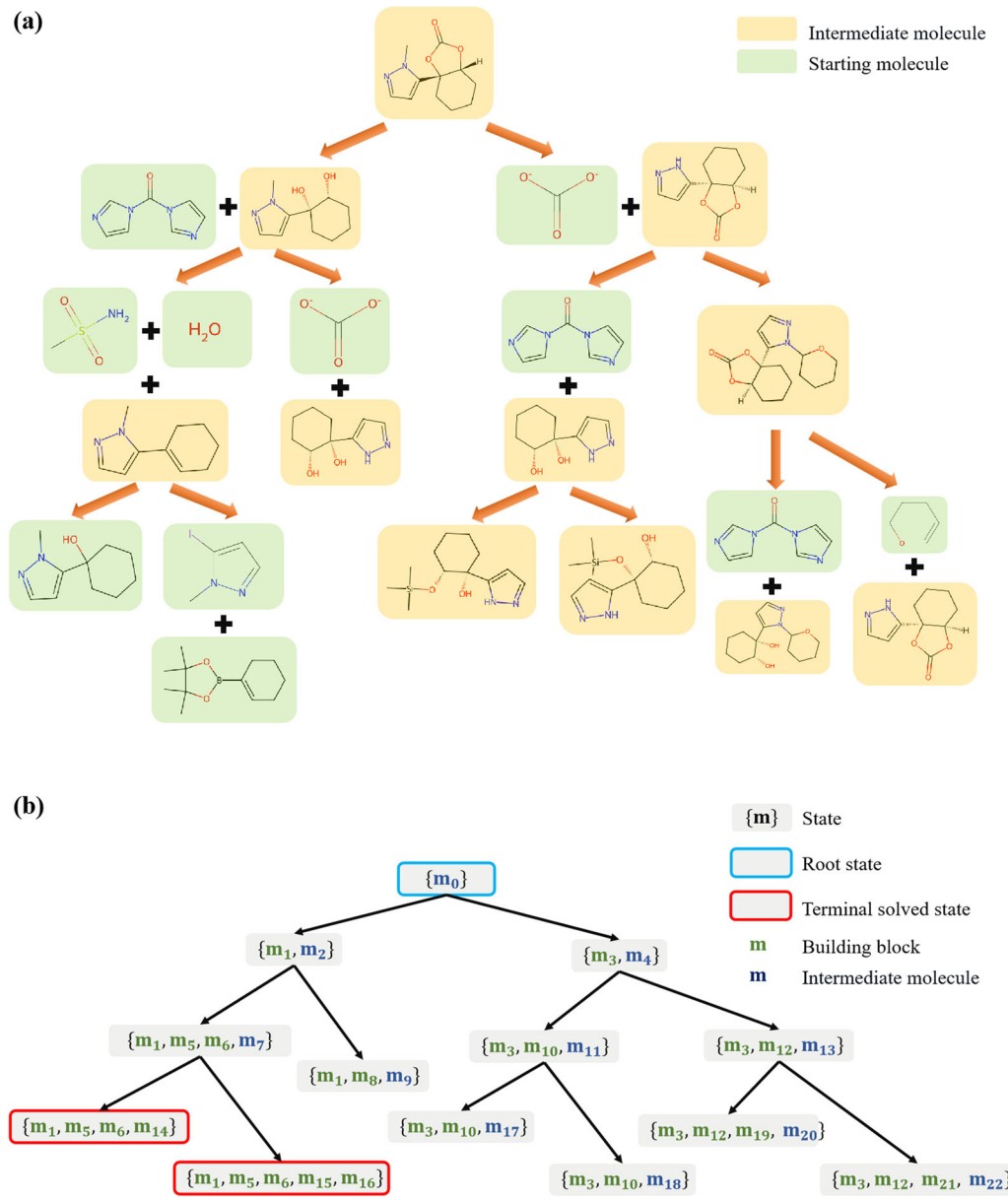

**Fig. 1 | Translation of the chemical retrosynthetic route representation to the search tree representation. a** Is the chemical representation of the synthesis plan, and **b** is the search tree representation. In the search tree, the states of the nodes encompass a set of molecules essential for the synthesis of the target molecule, including all unsynthesized intermediate molecules as well as the building blocks. The edges represent chemical reactions applied to the parent node.

proposed employing neural network-guided MCTS to do retrosynthetic planning, which almost reached the level of the literature routes in a double-blind AB test. There exists a trade-off between exploitation and exploration in MCTS due to the traversing tree policy[24]. The selected leaf node is evaluated for its expected subsequent cost, which is propagated back to all nodes on the traversed path to influence the subsequent search process. EG-MCTS[26] and GRASP[27] are also built on MCTS. Depth-first proof-number search (DFPN) has been utilized to address retrosynthetic planning[28,29]. Besides, A*-like search algorithms, including Retro*[30], Retro*+[31], GNNRetro[32], RetroGraph[33] and PDVN[34], have demonstrated promising results.Several widely used retrosynthetic planning software tools are publicly available, such as AiZynthFinder[35] and SynRoute[36]. DFPN is considered inferior to the other two search algorithms. MCTS and Retro* have demonstrated comparable search speeds and capabilities in route discovery. When comes to the route quality and diversity, MCTS outperforms Retro* in the 10,000 test molecules in PaRoutes[37,38]. However, the routes with a depth of more than 10 reactions have been excluded in PaRoutes[37], and

MCTS exhibits poor performance on these discarded molecules with longer paths, as will be discussed later. This selection process may introduce a bias in favor of the MCTS search. Actually, 15.3% of the routes in the USPTO test benchmark[30] exceed a length of 10. Despite the optimality of A* search being theoretically guaranteed[39], obtaining an optimal solution more efficiently remains a challenging problem in practice. Compared to MCTS, the search process of A* lacks exploratory behavior, and the estimated values are not updated once determined as more information is available[40].

Both A* search and MCTS used in retrosynthetic planning algorithms are heavily dependent on the quality of the heuristic guiding functions, including the single-step expansion policy and the cost estimator. Guided by biased heuristic functions, A* search may spend a considerable amount of effort expanding states on a non-optimal branch due to the lack of exploration, which significantly reduces the efficiency of the search process, as displayed in Fig. 2a. The search process of MCTS relies on a balance between exploitation and exploration. Insufficient exploration may lead MCTS to exhibit behavior similar to that of A* search. If exploration is over-

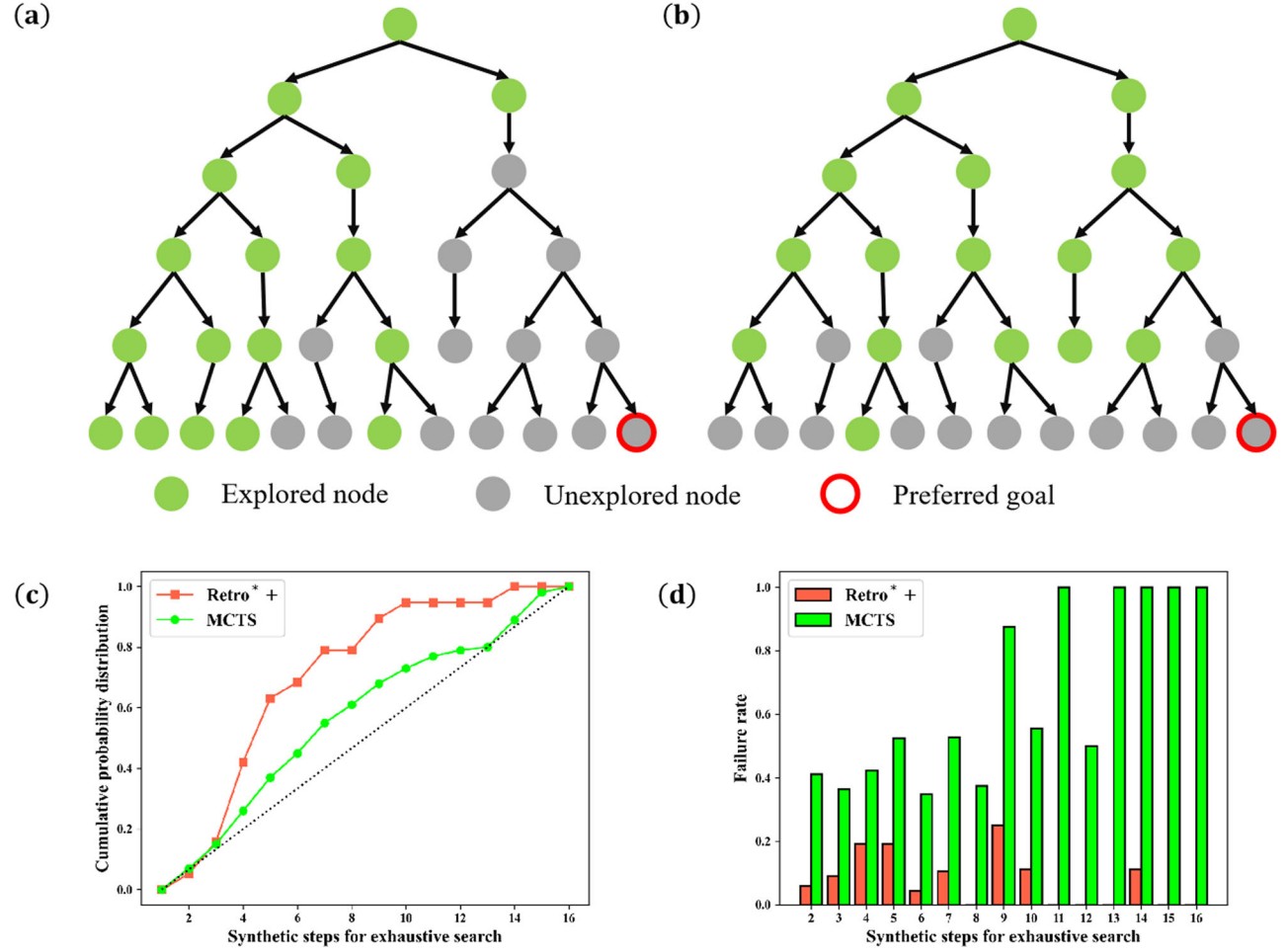

**Fig. 2 | Demonstration of the results by methods of insufficient or excessive exploration under the limited search time. a** The first failure case expands states on a non-optimal branch due to the lack of exploration; **b** The second failure case compulsively explores unnecessary branches and is unable to delve deeper toward the optimal solution; **c** The cumulative proportions of lengths of synthetic routes required by the exhaustive search for molecules that are failed to be synthesized by the search algorithm; **d** The failure rates of molecules with different path lengths to be synthesized by the exhaustive search (The source data for **c** and **d** is provided in Supplementary Data 1 and 2). Retro*+ and MCTS employ the same expansion function and value estimator. (MCTS does not utilize the pruning technique).

emphasized, a significant amount of effort will be spent to explore unnecessary branches compulsively. Pruning can be employed during the search process to reduce those meaningless explorations, but it may also result in the exclusion of the optimal solution due to insufficient simulations in the early search stages. Taking retrosynthetic planning results of the USPTO test set as an example, feasible synthetic pathways are provided alongside the target molecules[30], which are obtained by exhaustive search with reactions in the USPTO database. As shown in Fig. 2c, considering the failed molecules by Retro*+[31], which is an A*-like search algorithm, 78.95% of them can be synthesized within 7 steps in the USPTO benchmark, and 89.47% can be synthesized within 9 steps. This observation suggests that A* search may delve too deeply into unproductive branches, and fail to find the solution under a limited search time. In contrast, 45.0% failure cases of MCTS require at least 8 steps, and the failure rate of MCTS for molecules that require more than 8 synthetic steps by exhaustive search is 84.8%. The compulsive exploration of MCTS prevents it from proceeding deeper toward the optimal solution. Compared to the A* search, MCTS encounters difficulties in synthesizing complex molecules with longer routes within the same search time.

In general, A* search is guaranteed to find the optimal solution, but the efficiency of A* in problem-solving is compromised due to the deficiency in exploratory. Although MCTS is capable of exploration, compulsive exploration of meaningless states also decreases the efficiency of the search process. To address the aforementioned limitations, the MCTS exploration enhanced A* search algorithm (MEEA*) is proposed to incorporate the

exploration capacity of MCTS into A* search. Experiments are conducted on ten organic molecule datasets, including the widely used USPTO benchmark[30]. To the best of our knowledge, MEEA* achieves a 100.0% success rate on the USPTO benchmark for the first time, and improves the success rate on all 11,310 test molecules of the ten datasets from 60.50% to 65.14%, demonstrating the effectiveness of the proposed search algorithm. What's more, path consistency (PC) constraint is considered to improve the generalization ability of the cost estimator. The PC-enhanced version of MEEA* significantly improves the overall synthesis success rate to 76.27%. As is known, biosynthetic natural products (NPs) and the aforementioned tested organic molecules are derived from two distinct structural spaces, and thus the NP synthesis may be beyond the capabilities of the existing methods for the above organic molecule benchmarks[25,41]. Here, MEEA* is easily extended to work on NP synthesis by adopting the single-step expansion policy from BioNavi-NP[41] and obtains the NP synthesis success rate at 97.68%, which is much higher than 90.2% by the state-of-the-art BioNavi-NP.

## Results
### MEEA* search algorithm
MCTS comprises two crucial operations: the pUCT tree policy[42] for selecting the expanded leaf node, which enables a balance between exploration and exploitation, and the revision of state values during the search process, which influences the selection of nodes to be expanded in subsequent iterations. Correspondingly, the node with the smallest $f$-value is

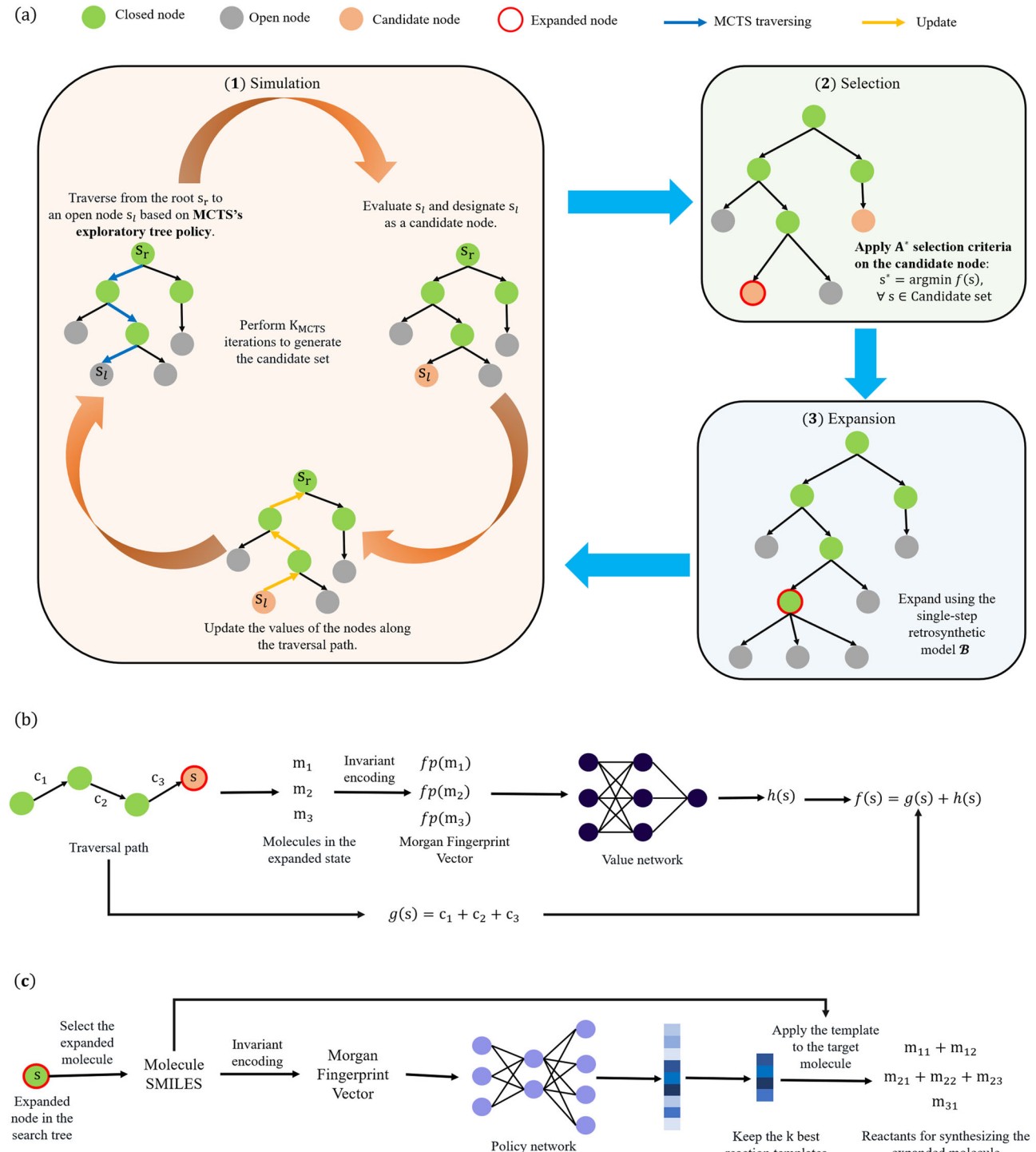

**Fig. 3 | The overall framework of MEEA* retrosynthetic planning search algorithm. a** The search process of our MEEA* algorithm includes three steps. (1) Simulation: conduct $K_{MCTS}$ iterations to generate the candidate set from the open set; (2) Selection: select the state with the minimum $f$ value from the candidate set; (3) Expansion: expand the selected state using the single-step retrosynthetic model $\mathcal{B}$. **b** Evaluation of states during the Simulation step of MEEA*. $g(s)$ is the summation of reaction costs along the traversal path, and $h(s)$ is estimated by the value network, considering all molecules in the state. **c** Single-step retrosynthetic model $\mathcal{B}$ used in the Expansion step. The first non-building block molecule is selected for expansion. The top $k$ best reaction templates are obtained based on the priors provided by the policy network, which are applied to the expanded molecule to generate its possible precursors.

expanded in the A* search algorithm, in which $f$ value is the summation of $g$ value, the accumulated cost from the initial state $s_0$ to $s_t$, and $h$ value, the expected cost from $s_t$ to the preferred goal $s_G$,

$$f(s_t) = g(s_t) + h(s_t). \tag{1}$$

MEEA* search algorithm is proposed by integrating the MCTS with A* search. As depicted in Fig. 3a, the search process of our MEEA* consists of three steps:

- Simulation: A set of candidate nodes is collected by performing $K_{MCTS}$ MCTS simulations without node expansion. The pUCT tree policy is employed to traverse to leaf nodes, thereby introducing exploratory

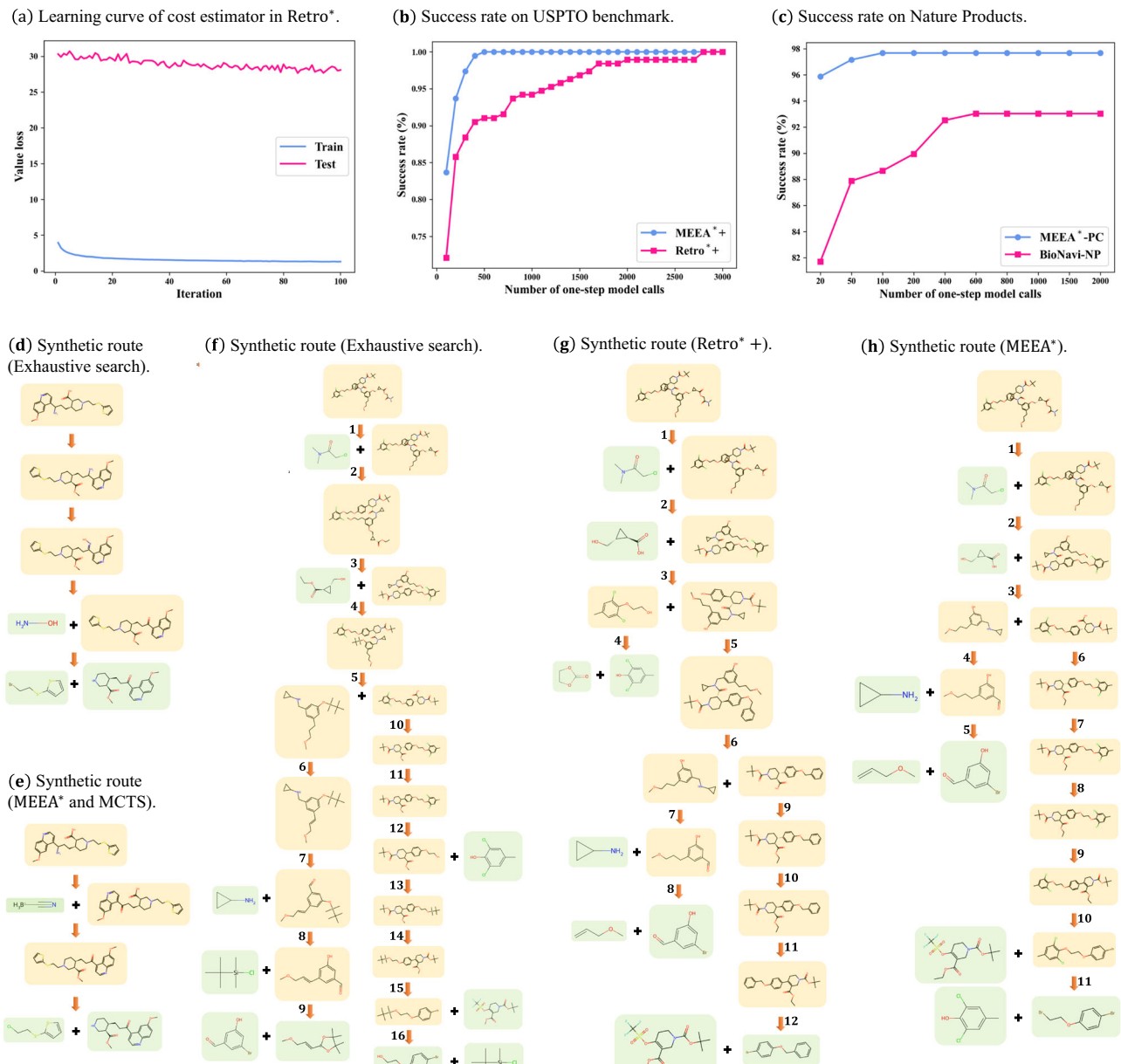

(a) Learning curve of cost estimator in Retro*.

(b) Success rate on USPTO benchmark.

(c) Success rate on Nature Products.

(**d**) Synthetic route (Exhaustive search).

(**f**) Synthetic route (Exhaustive search).

(**g**) Synthetic route (Retro* +).

(**h**) Synthetic route (MEEA*).

(**e**) Synthetic route (MEEA* and MCTS).

**Fig. 4 | MEEA* achieves superior performance compared with A* search and MCTS. a** The learning curve of the cost estimator in Retro* illustrates the overfitting problem; **b** Success rate on USPTO benchmark with different iterations. Guided by the same heuristic function, Retro*+ also achieves 100% success rate after enough iterations, but MEEA* is much more efficient; **c** Success rate on Natural Products (The source data for **a–c** is provided in Supplementary Data 3–5); **d, e** Synthetic route for molecule COc1ccc2nccc(C(N)CC[C@@H] 3CCN(CCSc4cccs4)C[C@@H]3C(=O)O)c2c1. MEEA* provides a shorter reaction pathway than the exhaustive search. Retro*+ fails to provide a solution, although the target can be synthesized in three steps. MCTS yields the same solution with MEEA*, but MCTS requires 35 expansions while MEEA* only requires 11 expansions. **f–h** Synthetic route for molecule COCCCc1cc(CN(C(=O)[C@H]2CN(C(=O) OC(C)(C)C)CC[C@@H]2c2ccc(OCCOc3c(Cl) cc(C)cc3Cl)cc2)C2CC2) cc(OC[C@@H]2C[C@H]2C(=O)OCC(=O)N(C)C)c1. Reaction pathways provided by the exhaustive search, Retro*+, and MEEA* with lengths of 16, 12, and 11 respectively. MCTS fails to provide a solution. (MEEA*, MCTS, and Retro*+ are guided with the same single-step expansion policy and cost estimator).

information into the candidate set. $f$ values of the traversed nodes are estimated using the cost estimator. The evaluation of the leaf node is updated to all nodes in the path from the root $s_r$ to the leaf $s_l$ in a backward pass.

- Selection: The node $s$ with the smallest $f$ value in the candidate set is identified as the next node in the search tree to be expanded.
- Expansion: Integrate the children of $s$ into the search tree provided by the single-step retrosynthetic model.

MEEA* is a heuristic search algorithm. As displayed in Fig. 3b, c, a single-step retrosynthetic model is employed to provide possible synthetic reactions. The policy network takes as input the Morgan fingerprint of the first non-building block molecule in the expanded state. The top $k$ reaction templates are utilized to produce potential precursors of the target molecule. According to Eq. (1), the $f$ value of state $s$ is evaluated by calculating $g(s)$ based on the traversal path and using a value neural network as a cost estimator to predict $h(s)$. The details of MEEA* are presented in the Method section. Compared to the A* search, the existence of a candidate set injects the exploratory capability of MCTS into MEEA*. The simulation times $K_{MCTS}$ is a hyperparameter balancing the influence of A* search and MCTS on MEEA*. Besides, MEEA* also preserves the optimality of A* search, because all nodes expanded by A* search are also expanded by MEEA* under the

**Table 1 | Performance summary on USPTO benchmark with a hard time limit of 500 single-step model calls, or 10 min**

| Algorithm | Success rate | Avg length | Avg cost | Avg time |
|---|---|---|---|---|
| Exhaustive search | 100.0% | 6.67 | 10.66 | – |
| Retro[*30] | 86.84% | 9.71 | 15.33 | 157.11 |
| Retro[*-030] | 79.47% | 11.21 | 19.40 | 208.09 |
| Retro[*+31] | 91.05% | 8.74 | 15.23 | 100.15 |
| Retro[*+-031] | 96.32% | 7.69 | 11.66 | 96.22 |
| RetroGraph[33] | 99.47% | 6.33 | 12.92 | **45.13** |
| MCTS[a25] | 33.68% | >21.22 | >44.91 | 370.51 |
| MCTS+[a25] | 35.79% | >20.55 | >43.48 | 365.21 |
| GRASP[27] | 98.94% | 6.17 | 13.91 | 48.47 |
| MEEA[*] | **100.0%** | **6.11** | **9.88** | 55.72 |

Molecules that fail to synthesize are assigned a large synthetic length and cost. The minimum average length and cost are calculated by setting the length and cost of successfully synthesized molecules to 0.

Values marked with bold are the best performance under each metric.

[a]The success rates of MCTS and MCTS+ are borrowed from[30].

**Table 2 | The success rate comparison tested in 10 datasets**

| Dataset | Retro[*30] | Retro[*+31] | EG-MCTS[26] | MEEA[*] | MEEA[*]-PC |
|---|---|---|---|---|---|
| USPTO[30] | 86.84% | 91.05% | 96.84% | **100.0%** | 94.74% |
| logS[44] | 67.08% | 69.29% | 71.74% | 73.22% | **80.34%** |
| BBBP[48] | 47.87% | 52.46% | 54.92% | 57.70% | **66.88%** |
| ClinTox[52] | 38.69% | 43.15% | 45.54% | 50.00% | **60.51%** |
| logP[45] | 53.96% | 61.14% | 62.72% | 65.24% | **73.72%** |
| DPP4[49] | 68.52% | 78.59% | 77.05% | 83.63% | **96.04%** |
| BACE[50] | 33.71% | 38.35% | 40.07% | 40.15% | **56.32%** |
| Ames[47] | 57.40% | 63.51% | 66.61% | 68.29% | **78.74%** |
| Toxicity LD50[46] | 55.39% | 59.98% | 64.22% | 66.28% | **72.94%** |
| SVS[51] | 50.14% | 55.93% | 58.23% | 60.19% | **73.01%** |
| Total | 54.18% | 60.50% | 62.20% | 65.14% | **76.27%** |

Values marked with bold are the best performance under each metric.

consistency assumption of the heuristic value $h$ predicted by the neural network[39]. The detailed proof is provided in Supplementary Note 1.

**Path consistency enhanced cost estimator**

Reliable guiding functions benefit the search process of MEEA[*] greatly. From a chemical perspective, the synthetic difficulty of a molecule is related to its structural complexity. The high structural diversity and sparse training data present a challenge in accurately estimating their synthetic cost. What's more, the synthetic cost is also influenced by the available building blocks. Complex molecules can be synthesized expeditiously in a few steps if precursors are readily available[25]. The synthetic cost of molecules with similar structures can vary considerably, due to differences in the availability of their precursors. Therefore, the generalization ability of the cost estimator is critical to the success of the retrosynthesis. As shown in Fig. 4a, Retro[*30] exhibits a substantial gap between its testing and training error of their cost estimator, indicating a severe overfitting problem.

In this paper, MEEA[*]-PC is proposed to alleviate this problem. Path consistency (PC), which is $f$ values on one optimal path should be identical, is suggested to be used as a constraint to improve the learning efficiency[40]. PCZero[43] incorporated PC into MCTS, and experiment results imply that PC sacrifices accuracy on the training set slightly to improve the generalization ability on the testing set greatly. In this paper, PC is utilized to train a more reliable cost estimator to assist MEEA[*] search by considering PC as a regularization term $\mathcal{L}_{PC}$ of the regular loss function $\mathcal{L}_{RL}$, i.e.,

$$\mathcal{L}(\theta) = \mathcal{L}_{RL}(\theta) + \lambda \mathcal{L}_{PC}(\theta), \quad (2)$$

where $\lambda$ is a hyperparameter. PC is realized by minimizing the deviation of the estimated $f$ values to the mean $f$ value of all nodes in the synthetic route. To train the PC-enhanced cost estimator via reinforcement learning, we construct a training set that consists of the synthesis pathways of molecules in the USPTO training set. The synthesis ways are identified by MEEA[*] with the heuristic functions provided in Retro[*+31].

**Retrosynthetic evaluation of MEEA[*] search algorithm**

Experiments are conducted to validate the effectiveness of our MEEA[*] search algorithm. Following the literature[26,27,30,31,33], commercially available molecules in eMolecules are used as the building block set. Firstly, experiments are conducted on the widely used USPTO benchmark, containing 190 molecules. Following previous works[26,27,30,31,33], all algorithms are limited to a maximum of 500 single-step model calls, or 10 min of real-time. Details of benchmark algorithms are provided in Supplementary Note 2 and Supplementary Table 1. It needs to be noted that although MEEA[*] employs

MCTS to conduct lookahead search, MCTS simulation in MEEA[*] does not call single-step models. The invocation of the one-step retrosynthetic model accounts for the majority of the runtime. Average times are evaluated by the number of calls to the single-step expansion policy to measure the efficiency of algorithms. For molecules that appeared in the USPTO dataset, an exhaustive search is employed to collect all possible pathways for synthesizing the target molecule utilizing chemical reactions from the USPTO dataset, and the shortest route is selected as the reference synthetic route. The performance of an exhaustive search is used as a benchmark for evaluation. As presented in Table 1, MEEA[*] achieves a remarkable 100.0% success rate, outperforming the existing A[*]-based and MCTS-based algorithms. Additionally, the synthesized routes demonstrate superior quality compared to the existing algorithms, in terms of both route length and synthesis cost. The substantial performance improvement highlights the effectiveness of our MEEA[*]. Although MEEA[*] requires the use of MCTS as a pre-search to determine the candidate set, the execution time spent for each expansion slightly increases. However, because of MEEA[*]'s ability to find solutions with fewer expansions, the overall runtime is actually shorter. Taking the USPTO test benchmark as an example, Retro[*]+ spends 6578 s while MEEA[*] only spends 3416 s. RetroGraph[33] achieves the shortest search time by adopting batch processing, which enables the expansion of nodes to be shared during the synthesis of multiple molecules. Theoretically, Retro[*]+ can also identify synthesis pathways for all molecules in the USPTO benchmark since it is guaranteed to find the optimal solution if the search resource is unlimited. As shown in Fig. 4b, the success rate of Retro[*]+ achieves 100.0% when the number of one-step model calls is 2800, which is more than five times the number by MEEA[*]. There is a significant improvement in search efficiency achieved by MEEA[*]. The success rate on the USPTO benchmark with different single-step model calls is displayed in Supplementary Note 5 and Supplementary Table 3.

The USPTO benchmark is widely utilized to evaluate retrosynthetic planning algorithms. However, molecules in the test benchmark need to satisfy two conditions. A synthetic route with reactions in the USPTO database is available, and each reaction in the synthetic route must rank within the top 50 predictions of the single-step expansion model of Retro[*30]. These two screening criteria have intentionally reduced the difficulty of this benchmark. The post-filtered test dataset exhibits inherent biases, thereby inadequately representing the real distribution of molecules. Therefore, nine additional datasets are included to ensure the authenticity of the evaluations, which also have strong practical significance. The distribution of molecules in the USPTO benchmark and the other nine molecules is presented in Supplementary Note 10 and Supplementary Fig. 3. More detailed information is listed in the Method. The USPTO benchmark requires that each molecule in the test set possesses at least one synthetic pathway, and the

**Table 3 | The results by imposing a time limits of 10 min on every run on USPTO benchmark**

|  | Retro*+ | A*($K_{MCTS} \to \infty$) | MCTS($K_{MCTS}$ = 1) | MEEA* |
|---|---|---|---|---|
| Success rate | 96.84% | 99.47% | 97.37% | 100.0% |
| First solution time | 43.67 s | 22.99 s | 55.33 s | 19.93 s |
| Number of clusters | /[a] | 4.09 | 4.26 | 4.17 |
| TED | 11.10 | 10.15 | 11.27 | 10.94 |
| Leaves overlap | 0.606 | 0.638 | 0.582 | 0.605 |

[a]Retro*+ is not designed to generate multiple synthetic pathways, and only the best route is provided. The number of clusters for Retro*+ is not presented in the table.

pathway is constructed by the reactions ranking within the top 50 predictions of the single-step expansion model. The molecules in the nine additional datasets are not subject to these restrictions, and may not have feasible synthesis routes among the top 50 chemical reactions, which renders them more representative of real-world molecule distributions and raises higher requirements for the single-step expansion policy. As shown in Table 2, MEEA* outperforms Retro*+ and EG-MCTS in all ten datasets with 65.14% overall success rates. This clearly demonstrates the effectiveness of MEEA* on a large-scale test set. The results of RetroGraph[33] and GRASP[27] are not presented as their codes are not publicly available. On the other hand, MEEA* focuses on improving the search algorithm. While Retro*+ and MEEA* utilize the same heuristic function, the significant performance improvement provides compelling evidence of the effectiveness of our search algorithm. Investigation on the hyperparameter $K_{MCTS}$ is displayed in Supplementary Note 8 and Supplementary Fig. 1.

### Trade-off between quality and diversity

Existing works usually conducted experiments by limiting the number of calls of the single-step model[26,27,30,31,33]. Varying the single-step model can result in uneven resource allocation across different models, and limiting search with wall-clock time is recommended[37,38]. Besides the success rate, the diversity of the generated routes is highly recommended as a criterion[37,38]. For a given molecule, multiple diverse synthetic pathways are desired to be provided to human experts for further selection. The generated routes are clustered into different groups and the number of clusters is regarded as a metric of diversity[37]. Tree edit distance (TED) is calculated to measure the similarity between the algorithm-generated synthetic pathway and the referenced pathway, and the leaves overlap is computed as the average ratio of the building blocks present in both the generated route and the reference route. The synthetic routes provided in the USPTO benchmark are used as the referenced routes to compute TED and the leaves overlap. During the experiment, at most 25 synthetic routes are generated to compute the number of clusters, considering the computational time. The TED value and the leaves overlap are calculated based on the first generated route. A* search and MCTS are included for comparisons, and they are implemented by setting $K_{MCTS} \to \infty$ or $K_{MCTS}$ = 1 in MEEA*. Unsolved molecules are excluded from the calculation of the evaluation metrics.

The results of the success rates and the first solution time are reported in Table 3, where the first solution time means the time used to find the first feasible solution. MCTS typically takes the longest time to find the first feasible solution, due to its compulsory exploration to unnecessary nodes. MEEA* not only solves all molecules with 100.0% success rate, but also only requires 19.93 seconds on average, significantly less than the 55.33s for MCTS and 22.99s for A*. The results again demonstrate the high efficiency of the proposed MEEA*. It is observed that the above results are basically consistent with the ones by limiting the number of calls to the single-step retrosynthetic model. The reasons are as follows. The invocation of the single-step retrosynthetic

model accounts for the majority of the runtime. When the single-step model is kept fixed, capping the number of model calls is a reliable setting to evaluate the performances of retrosynthetic planning methods. Since the neural network architecture of the expansion policy of MEEA* remains the same as that of Retro* and Retro*+, the computational resources consumed by each single-step model invocation are basically identical. Moreover, considering the small variations in implementing various search algorithms, there exist only small differences between the runtime and the number of calls to the single-step retrosynthetic model. The success rate with different real search time is displayed in Supplementary Note 9, including both Supplementary Table 6 and Supplementary Fig. 2.

According to results in Table 3, the routes provided by MCTS have greater diversity with larger cluster numbers. Conversely, the routes generated by A* are more closely aligned with the routes of exhaustive search with a smaller TED and a higher proportion of overlapping leaves. This result implies a deficiency in exploratory because the routes of exhaustive search are used to train the heuristic guidance. MEEA* is a hybrid algorithm that combines the characteristics of A* and MCTS, and it achieves a good balance between route quality and diversity.

### Evaluation of path consistency enhanced MEEA*

As demonstrated in the previous sections, MEEA* borrows the single-step expansion policy and cost estimator from Retro*+, and achieves better performance than Retro*+. Here, the performance can be further improved by considering a better policy or cost estimator as follows. On the one hand, the cost estimator from Retro*+ predicts the synthetic cost of only one molecule at a time and calculates the total cost of the current state as the summation of predicted costs for all molecules related to the state. A cost estimation network is proposed. It is capable of taking all molecules within the current state jointly as input and computing the state cost in a direct and more accurate manner. The cost network is trained by minimizing the mean square error of cost prediction, and PC is employed to improve the generalization capability via Eq. (2). Expansion policy is also updated on the newly generated data by MEEA*

With the guidance of the updated policy and the cost estimator trained under the PC constraint, the overall success rate of MEEA* has significantly improved from 65.14% to 76.27%. Note that, only the success rate of USPTO decreased, while the performance improved across all other nine datasets. This observation is possibly due to the fact that there exists a distribution bias in the USPTO test benchmark due to expansion policy-based molecule filtering. MEEA* is a search algorithm, which can be combined with different single-step expansion models. Employing a better single-step expansion model can also enhance the performance of MEEA*. More experimental results are presented in Supplementary Note 7 and Supplementary Table 5.

### Synthesis of natural products

Natural products (NPs) play a significant role in drug discovery since 60% of FDA-approved small molecule drugs are either NPs or their derivatives. However, synthetic pathways of over 90% of NPs are unknown. Biosynthetic NPs and the aforementioned tested organic molecules are derived from two distinct structural spaces[41]. Most existing organic retrosynthesis tools cannot be directly applied to the synthesis of NPs[25]. Recently, BioNavi-NP[41] specifically built an expansion policy for natural products, which was combined with Retro*[30] to trace NPs back to biologically plausible building blocks. When limiting the search algorithm to 100 single-step model calls and 50 expansions, the success rate is improved from 88.66% to 97.68% by replacing Retro* with our MEEA*-PC. As shown in Fig. 4c, if increasing the iterations to 2000, the success rate of Retro* stays at 93.04%, which remains inferior to our MEEA*. The search space for both BioNavi-NP and MEEA* is identical as the same expansion policy is employed. BioNavi-NP may be trapped in the non-optimal branches due to the lack of enough exploration. The number of search iterations required to overcome this limitation in BioNavi-NP will significantly surpass that of MEEA*. Overall, MEEA* has

achieved significant performance improvements in both general organic molecules and natural molecules. An example of the synthesis of a natural product is given in Supplementary Note 11 and Supplementary Fig. 4.

## Case study

Figure 4d, e displays the synthetic pathway of a molecule that can be synthesized with four chemical reactions by the exhaustive search. Both MEEA$^*$ and MCTS have discovered the same synthetic pathway, which exhibits a shorter route length. MCTS requires 35 single-step model calls while MEEA$^*$ only requires 11 calls, indicating that the compulsory exploration of MCTS results in lower efficiency compared to MEEA$^*$. Guided by the same heuristic functions, Retro$^*$+ is trapped in the first failure case depicted in Fig. 2a and fails to identify the synthetic route, although the target molecule can be synthesized with four steps.

Figure 4f–h illustrates the synthesis planning of a complex molecule. The exhaustive search, Retro$^*$+, and MEEA$^*$ provide reaction pathways with lengths of 16, 12, and 11, respectively. MCTS fails to find a solution within the given time limit. The compulsory exploration in MCTS prevents it from effectively delving into deeper regions in the search tree within the given time limit. The search process is trapped in the second failure case displayed in Fig. 2b. Furthermore, due to the exploratory property, MEEA$^*$ requires a slightly higher number of single-step model calls for finding the synthetic pathway than Retro$^*$+, but it successfully identifies a shorter synthetic route. The two retrosynthesis cases above have demonstrated the superiority of MEEA$^*$ from different perspectives.

Besides, MEEA$^*$ is applied to complex drug molecules to illustrate its practical significance, including Paxlovid, Fostemsavir, Enarodustat, Pacritinib, and Oteseconazole. MEEA$^*$ has successfully identified the synthesis pathway for the above drugs, and the synthetic plans are illustrated in Supplementary Note 12 and Supplementary Fig. 5.

## Discussion

In this paper, we proposed an efficient retrosynthetic planning algorithm MEEA$^*$, incorporating the exploratory behavior of MCTS into A$^*$ search by providing a lookahead search. The quality of the guiding function is crucial for the success of heuristic search. A$^*$ search is prone to being misled into non-optimal branches due to the lack of exploration ability. The compulsive exploration in MCTS limits the exploration depth within a finite iteration, rendering it less effective than A$^*$ in synthesizing complex molecules. Compared to A$^*$, MEEA$^*$ escapes local optimal branches by incorporating exploration. In contrast to MCTS, MEEA$^*$ prevents exhaustive exploration of all branches by utilizing A$^*$'s efficient node expansion criteria. To the best of our knowledge, MEEA$^*$ achieves 100.0% success rate on the widely used USPTO benchmark for the first time. Guided with the same heuristic functions with Retro$^*$+, MEEA$^*$ improves the success rate on all 11,310 molecules from 60.50% to 65.14%. Path consistency is considered to improve the generalization capacity of the cost estimator and significantly improves the overall synthesis success rate to 76.27%. Experiments on natural products are conducted. Guided with the single-step expansion policy provided by BioNavi-NP, the success rate of synthesizing NPs has been increased to 97.68% from BioNavi-NP's 90.2%. In summary, MEEA$^*$ has achieved significant performance improvements in both general organic molecules and natural molecules. Besides, we have provided a theoretical guarantee that MEEA$^*$ can definitely find the optimal solution.

There still remain certain challenges for the refinement of the MEEA$^*$. For example, the chemical reactions suggested by the single-step retrosynthesis model may present significant barriers to implementation in practice. Our work mainly focuses on the multi-step synthesis search algorithm, and the single-step model primarily draws from previous works. It is recommended to utilize a more reliable single-step retrosynthesis model or to implement additional filtering of the generated potential chemical reactions. Additionally, MEEA$^*$ integrates the exploratory nature of MCTS with A$^*$ search. It is critical to strike a balance between the influence of MCTS and A$^*$ by selecting an appropriate value for $K_{MCTS}$. The necessity of

exploration in MEEA$^*$ is dependent upon the reliability of the heuristic function. If the heuristic guidance is highly reliable, exploration will be necessary because the optimal solution can be identified through a best-first search directly. In such cases, MEEA$^*$ should use a larger value of $K_{MCTS}$ to prioritize exploitation and move closer to A$^*$ search. When the heuristic functions are unreliable, the importance of exploration becomes more crucial, MEEA$^*$ should choose a smaller value of $K_{MCTS}$ to better leverage the power of MCTS. What's more, molecules are represented by one-dimensional vectors, which may not capture sufficient stereochemical information. Graph neural networks offer a promising approach to leveraging more comprehensive molecular information.

## Methods

### Markov decision process for retrosynthesis planning

The Markov decision process (MDP) is a mathematical framework used to model sequential decision-making problems for an agent operating within an environment. It is described with state space $\mathcal{S}$, action space $\mathcal{A}$, transition function $\mathcal{T}$ and cost $\mathcal{C}$. The objective is to identify a pathway for synthesizing the target molecule $m_0$ using building blocks from $\mathcal{I}$. In this paper, we formulate a retrosynthesis planning problem as an MDP as follows:

- State space $\mathcal{S}$: A state $s \in \mathcal{S}$ is defined as a set of molecules $\{m_i, m_j, \cdots \}$ required for the synthesis of the target molecule. The initial state is $s_0 = \{m_0\}$, consisting only of the target molecule. The retrosynthesis of $m_0$ is considered successful if a state $s_t \subseteq \mathcal{I}$ is identified. Molecules in a state are sorted based on their SMILES representation.

- Action space $\mathcal{A}$: An action $a \in \mathcal{A}$ represents a potential chemical reaction that can be utilized to decompose a molecule into its constituent reactants. The action space is vast due to the multitude of chemical reactions available for synthesizing a molecule. The single-step retrosynthetic model $\mathcal{B}$ selects the $k$ most promising chemical reactions as legal actions i.e., $B(m) = \left\{ a_i, \pi(a_i|m) \right\}_{i=1}^{k}$, where $\pi(a_i|m)$ is the associated probability of taking action $a_i$ for molecule $m$. The remaining reactions are discarded to narrow down the width of the search tree.

- Transition model $\mathcal{T}$: When taking action $a_t$ in state $s_t$, $\mathcal{T}(s_t, a_t)$ determines the reaching state $s_{t+1}$. It replaces the product molecule associated with action $a_t$ in state $s_t$ with its corresponding reactants, yielding the subsequent state $s_{t+1}$. Actions are always applied to the first molecule in the state. For example, suppose that the current state is $s_t = \{m_1, m_2\}$, and the action $a_t$ is the reaction $m_3 + m_4 \rightarrow m_1$. The resulting state $s_{t+1} = \mathcal{T}(s_t, a_t) = \{m_2, m_3, m_4\}$.

- Cost $\mathcal{C}$: Reaction cost $\mathcal{C}(s, a, s')$ returns the cost when transitioning to $s'$ via action $a$ in state $s$. In practice, $\mathcal{C}$ is defined as $-\log p_r(a|m)$, where $p_r(s|m)$ is a predictive model trained on a real reaction dataset to estimate the probability of selecting reaction $a$ for molecule $m$ in the physical world.

### MEEA$^*$ search

MEEA$^*$ search is a hybrid algorithm that introduces the exploration of MCTS into A$^*$ search. MCTS follows iteratively four steps:

- Selection: Traverse from the root node to a leaf node with an exploratory tree policy.
- Expansion: Incorporate promising children into the search tree based on the single-step retrosynthetic expansion policy.
- Evaluation: Score the leaf by either fast rollout or heuristic value models.
- Update: Back up the simulation results to all states on the traversed path. A$^*$ search maintains two state sets: the OPEN set for generated but unexplored nodes and the CLOSED set for explored nodes. Initially, the CLOSED set is empty and the OPEN set contains only the root state. The search process iteratively executes the following steps:
- Selection: If the OPEN set is empty, terminate the search with failure. Otherwise, identify the state $s$ with the smallest $f$ value in the OPEN set.

- Expansion: Generate the child states of $s$ based on the expansion policy. If $s_G$ is generated, terminate with success. Otherwise, mark state $s$ as CLOSED and add generated child states to the OPEN set. Besides the OPEN set and CLOSED set, MEEA$^*$ maintains an additional candidate set, which is collected with the exploratory MCTS. Each edge in the search tree stores the visit count $N(s, a)$, the prior probability $\pi(s|a)$ received from the expansion policy network, and the entire synthetic cost $Q(s, a)$. The transition function $s' = \mathcal{T}(s, a)$ is deterministic. The search process of our MEEA$^*$ consists of three steps: simulation, selection, and expansion. In the first phase of MEEA$^*$, simulations are conducted for $K_{MCTS}$ iterations without node expansion to collect a candidate set. Each simulation consists of three steps.

- Traverse from the root to a leaf node according to a tree policy. In order to further encourage exploration, a variant of pUCT is adopted, and a uniform distribution is employed as the prior policy. The next action is selected via:

$$a^* = \arg\max_{a \in \mathcal{A}} \left\{ -Q(s, a) + c_{puct} \frac{1}{|\mathcal{A}|} \frac{\sqrt{\sum_{b \in \mathcal{A}} N(s, b)}}{1 + N(s, a)} \right\}, \quad (3)$$

where $c_{puct}$ is a hyperparameter to control the exploration level of the simulation. The tree policy is applied until a leaf node $s_l$ is found.

- Put the leaf node $s_l$ into the candidate set and calculate the $f$ value of $s_l$, which is the summation of $g(s_l)$ and $h(s_l)$ according to Eq. (1). $g(s_l)$ is the total cost of all reactions from $s_0$ to $s_l$, and the subsequent accumulated cost $h(s_l)$ is predicted by the heuristic cost estimator.

- Update the stored statistics for nodes in the path from the root $s_0$ to the leaf $s_l$.

$$Q(s, a) \leftarrow \frac{N(s, a) \times Q(s, a) + f(s_l)}{N(s, a) + 1}, N(s, a) \leftarrow N(s, a) + 1 \quad (4)$$

After $K_{MCTS}$ simulations, the candidate set with exploratory is determined. During the selection step, the candidate node with the lowest $f$ value is chosen from the candidate set to be expanded in the next step. Single-step expansion model $\mathcal{B}$ provides top 50 successor states, which are directly added to the search tree. If a newly added state contains exclusively building blocks, the synthetic pathway for the target molecule is successfully identified. The above three steps are iteratively performed until the iteration step budget has been exhausted. The simulation times $K_{MCTS}$ is a hyperparameter balancing the influence of A$^*$ search and MCTS on MEEA$^*$. When $K_{MCTS}$ is relatively small, MEEA$^*$ approaches MCTS with increased exploration and degenerates into pure MCTS when $K_{MCTS} = 1$. In this case, only one candidate node is available for expansion, which is determined by the exploratory tree policy of MCTS. When $K_{MCTS}$ grows, more leaf nodes are included in the candidate set. If $K_{MCTS}$ is sufficiently large, all opening nodes will be chosen as candidate nodes because of the exploratory nature of MCTS. Consequently, MEEA$^*$ is degraded to A$^*$ search, where the node with the lowest $f$-value among all open nodes is selected for expansion. Single-step model $\mathcal{B}$ and cost estimator are borrowed from Retro$^*$+ to assist the search process of our MEEA$^*$. The parameter $K_{MCTS}$ and $c_{puct}$ is set to 100 and 4.0 respectively. Experiments are conducted on Tesla V100 GPUs and Intel(R) Xeon(R) Gold 6238R CPU with 512G memory. There is a wide range for $K_{MCTS}$ to make MEEA$^*$ perform well.

## Synthesis pathway preparation
MEEA$^*$ is a heuristic search algorithm that requires an expansion policy and cost estimator to assist the search process. Although the guiding function can be provided by previous works, it is still necessary to train our own networks for better performance. The USPTO training set provided by Retro$^{*30}$ does not include synthetic pathways for molecules, so we are required to generate these routes by ourselves. One choice is to employ an exhaustive search to identify the shortest synthetic pathway of the given molecule using the chemical reactions available in the USPTO dataset, the same as Retro$^{*30}$. This strategy is effective in training a model with relatively high performance from scratch, but limits the performance of the model, making it challenging to surpass the capabilities represented by the training set[42]. The other choice is to update the heuristic model with reinforcement learning, collecting synthetic routes with MEEA$^*$ to further update its heuristic guidance iteratively. Retro$^*$+[31] has illustrated the effectiveness of this self-improved learning process.

MEEA$^*$ search is utilized to identify the synthetic pathways for molecules in the USPTO training set. The expansion policy and cost estimator from Retro$^*$+[31] are employed as the initial heuristic functions to assist MEEA$^*$ search. There are 299202 molecules in the USPTO training set, and MEEA$^*$ has successfully identified synthetic pathways for 299024 of them. The chemical reactions that appear in the collected synthetic pathways will be used to update the expansion policy. The synthetic cost is calculated as $c(a) = \log p_r(a|m)$, prepared for the training of the cost estimator. $p_r$ is the expansion policy used by Retro$^{*30}$, which is trained using actual chemical reactions and can be used to measure the likelihood of a reaction taking place. It needs to be noted that the updated policy and newly trained value function are exclusively employed in MEEA$^*$-PC, and additional ablation studies are provided in the Supplementary Note 6 and Supplementary Table 4 for further analysis.

## Single-step retrosynthetic model
In this paper, a template-based single-step retrosynthesis model is adopted by treating the problem as a classification task based on reaction templates. The identical network architecture as Retro$^*$+ is employed, which is a neural network comprising a single hidden layer with 512 dimensions. The network parameters are updated using the cross-entropy loss, with Retro$^*$+'s model serving as the initial model for the training process. We utilize the Adam optimizer with a learning rate of 0.001, and the batch size is set to 1024. The architecture of the network is presented in Supplementary Note 3.

## Cost estimator and path consistency
The cost estimator is employed to predict the cost of synthesizing all molecules in the state $s = \{m_i, m_j, \cdots\}$. However, previous cost estimator algorithms, such as Retro$^*$ and Retro$^*$+, were limited to predicting the synthetic cost of one molecule at a time. The total synthetic cost for state $s$ is calculated as the summation of independent predictions of synthetic costs for all molecules within $s$,

$$h(s) = \sum_{m \in s} h(m), \quad (5)$$

where $h$ is a cost estimation function. In this paper, we have designed a network to estimate the synthetic cost jointly for all molecules within a given state $s$, as illustrated in Eq. (6). This computational approach provides greater flexibility than the method in Eq. (5), allowing the model to utilize inter-molecular information to improve the prediction performance,

$$h(s) = h(m_i, m_j, \cdots). \quad (6)$$

Multiple molecules are fed into the same network to extract their feature embeddings. The global embedding $e^g$ is computed as the sum of all molecules' embedding $e_i^v$, which can be viewed as a representation of the entire state. Global embedding $e^g$ is further fed into a fully connected layer to obtain the cost estimation of the state directly.

Through analysis of the collected synthetic pathways, the states of each step in the synthesis process, as well as the associated real synthetic cost $c(s)$, are obtained and utilized as training data for the cost estimator. The mean

square error is utilized as the loss function, as shown in Eq. (7). The Adam optimizer with a learning rate of 0.001 is employed to update the parameters, and the batch size is set to 256.

$$\mathcal{L}_{RL}(s) = (h(s) - c(s))^2 \qquad (7)$$

After obtaining a function model that can directly evaluate the entire state, we consider incorporating the path consistency constraint into the model's learning process to improve its generalization ability. Path consistency is defined as $f$ values on one optimal path should be identical. For a collected synthetic pathway $\{s_0, s_1, \cdots, s_N\}$, PC constraint is evaluated by the deviation of the state evaluation $f(s_t)$ from the learning target, which is the average state evaluation along the estimated optimal path.

$$\mathcal{L}_{PC}(s) = \left(f(s) - \bar{f}(s)\right)^2, \bar{f}(s) = \frac{1}{N+1}\sum_{i=0}^{N} f(s_i) \qquad (8)$$

As illustrated in Eq. (2), the PC loss is used as a regularization term for the reinforcement learning loss, which is adjusted by the hyperparameter $\lambda$. During training, $\lambda$ is set to 5.5.

### Test dataset

The United States Patent Office (USPTO) dataset is a publicly available reaction dataset, which consists of approximately 3.8 million reactions collected from patents granted between 1976 and September 2016. It is widely used for machine learning applications because of its size, diversity, and accessibility. USPTO dataset is prepossessed and split into train/validation/test sets by Retro[*30] as a benchmark dataset for retrosynthesis planning problems. For each molecule in the USPTO dataset, a brute-force search is conducted to determine if it can be synthesized using reactions within the USPTO training data. For each synthesizable molecule, the shortest-possible synthesis pathways are collected to train a single-step expansion policy, which is used to provide the most potential reactions for multi-step retrosynthesis planning algorithms. Only molecules, for which the reactions in their synthesis pathways are all covered by the top-50 predictions by the single-step expansion policy, are preserved in the test benchmark. As a result, the distribution of molecules in the USPTO test benchmark differs from the distribution of molecules in real life.

To enable a more rigorous evaluation of the algorithm's performance in real-world scenarios, nine additional real-world molecular datasets are also utilized as the test set. logS[44] logP[45], and Toxicity LD50[46] are used to predict the solubility, hydrophobic property, and toxicity of molecules separately. The Ames dataset[47] is a standardized collection of data used to assess the genetic toxicity of bacteria, i.e., whether it induces mutations. Human blood-brain barrier penetration (BBBP)[48] is a crucial property in the process of drug design. Dipeptidyl peptidase-4 (DPP4) inhibitors[49] are an essential drug in the treatment of type-2 diabetes mellitus. $\beta$-secretase 1 (BACE-1) inhibitors[50] are a class of drugs that target the BACE-1 enzyme, which plays a key role in the production of $\beta$-amyloid peptides in Alzheimer's disease. Inhibition of protein-protein interactions in SVS[51] is of great interest in drug design and discovery, because dysfunction of PPIs can lead to various diseases, including immunodeficiency, autoimmune disorders, allergy, drug addiction, and cancer. ClinTox[52] compromises FDA-approved drugs and compounds that have undergone clinical trials but failed due to toxicity-related issues. Those drug molecules in ClinTox are selected by experts for clinical trials in the early stages of drug development, which is suitable to evaluate the efficacy of retrosynthesis planning algorithms in drug design. More information about the test dataset is summarized in Supplementary Note 4 and Supplementary Table 2. To increase the challenge of the test, datasets are preprocessed as follows:

- Remove molecules present in either the USPTO database or the building block set.
- Remove molecules solved by a heuristic BFS planning algorithm within 50 steps.
- Remove molecules that can be solved by Retro[*] in one step.

Although biosynthetic natural products and chemically synthesized compounds share certain similarities, they are fundamentally distinct in terms of structural space and reaction types[41]. NPs exhibit a greater diversity and more complex structures, and existing organic retrosynthesis tools cannot be directly used for biosynthesis prediction. What's more, NPs can be synthesized from a dozen simple building blocks, in contrast to millions of building blocks required in the preceding organic dataset. 388 molecules collected by BioNavi-NP are used to test the performance in the synthesis of natural products.

### Data availability
All related data in this paper are public. The eMolecules dataset can be downloaded from http://downloads.emolecules.com/free/2023-12-01/. The USPTO benchmark can be downloaded from https://github.com/binghong-ml/retro_star. Other test molecule datasets are available in https://weilab.math.msu.edu/DataLibrary/2D/. We provide the source data underlying Fig. 2c, d, as well as Fig. 4a–c, in Supplementary Data 1–5.

### Code availability
The source code of MEEA[*] is available at https://github.com/CMACH508/MEEA.

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

## Acknowledgements

This work was supported by the National Natural Science Foundation of China (62172273), and Shanghai Municipal Science and Technology Major Project (2021SHZDZX0102).

## Author contributions

Dengwei Zhao proposed the research, conducted experiments, analyzed the data, and wrote the manuscript. Shikui Tu improved the manuscript and supervised the overall project. Lei Xu provided the key idea of the algorithm and improved the manuscript.

## Competing interests

The authors declare no competing interests.
