## [Peer Review File · Communications Chemistry]

Reviewers' comments:

Reviewer #1 (Remarks to the Author):

The authors propose MEEA*, an algorithm for retrosynthesis analysis that combines Retro* and MCTS. The manuscript is fairly well-written and the authors do a good job in convincing the reader that their new algorithm is worth using. I have a few comments and suggestions that should be attended to before the manuscript is ready for publication.

I encourage the authors to justify more carefully one of their stopping criteria for the searches, i.e. the number of calls to the single-step model. This seems to insert a bias against MCTS that the authors are aware of, i.e. that MCTS generally calls the single-step model more often. I think it would be better to just impose a time limit and then report search time as an evaluation criteria.

The authors mainly compare the algorithms using the number of solved targets and the length of the routes. I encourage the authors to look into other evaluation criteria as has been suggested in the literature (Genheden & Bjerrum, *Dig Discov* 2022; Maziarz et al. 2023 arxiv:2310.19796). The criteria that the author have chosen are insufficient evaluation criteria.

Line 64, in introduction claims "demonstrated more promising results" but both "Genheden & Bjerrum, *Dig Discov* 2022" and "Maziarz et al. 2023 arxiv:2310.19796" gives a more nuanced view of the relative merit of MCTS and Retro* and this should be mentioned in the introduction.

On page 8, the link to e-molecule stock is dead as the company does not provide the downloads from previous years.

Line 194, page 8 states "It is important to note that these molecules may not necessarily have feasible synthesis routes among the top 50 chemical reactions," but what exactly the author means by this needs to be clarified.

Reviewer #2 (Remarks to the Author):

This paper proposes an interesting hybrid search algorithm for retrosynthesis planning by combining Monte Carlo Tree Search (MCTS) and A* search. Comprehensive experiments on 10 datasets including USPTO show state-of-the-art performance. MEEA* achieves 100% success on USPTO and also works well for natural products. This work demonstrates a novel hybrid search algorithm that draws on the strengths of both MCTS and A*.

This paper shows potential for publication. The reviewer would like to highlight the following comments.

1. The introduction section can be improved by drawing comparisons to similar work in the field. While there are comparisons to existing work, it is the reviewer's opinion that currently the manuscript does not provide adequate references to existing methods.
2. Y-axis label in figure 2 is slightly confusing. Please look into this. Authors state that "As shown in Figure 2 (c), 89.27% failure molecules of Retro*+, which is an A* -like search algorithm, can be synthesized within 7 steps by the expert, indicating that A* search delve too deeply into unproductive branches to find the solution under limited search time." Can this threshold of 89.27% be indicated in figure 2 for clarity?
3. Can the authors comment on how sensitive is the performance to the MCTS simulation?
4. The main text does not provide any detailed descriptions of the used neural networks, although these are main components of the introduced workflow. Adding implementation details to the manuscript will be highly useful.
5. What is meant by "The performance of the expert is based on the pathways provided by the USPTO benchmark, which is the best synthetic route using all reactions in USPTO."? If the performance is based on the USPTO benchmark, then how is it computed?
6. It is stated that "However, the USPTO training set provided by Retro* [13] does not include synthetic pathways for molecules, so we will generate these routes using retrosynthetic planning algorithms, serving as the training set for the following task." Did the training data come from other retrosynthetic planning algorithms? Couldn't the pathways provided by the USPTO benchmark be used in this setting? This needs further clarification.
7. A general critique of the manuscript is that the sentences tend to be too long and repetitive. Reframing such long sentences will improve readability. Eg: "Despite the widespread utilization of the USPTO benchmark for evaluating retrosynthetic planning algorithms, the difficulty of this benchmark has been intentionally reduced by including only molecules synthesizable with reactions in the USPTO database, where each reaction must rank within the top 50 predictions of the singlestep expansion model of Retro*"
8. A typo was found in "It is capable of talking all molecules within the current state jointly

as input and computing the state cost in a direct and more accurate manner.”

Dear Editor and Reviewers,

Thank you for your insightful comments and suggestions regarding our manuscript. We greatly appreciate the opportunity to address the reviewer's comments and make the necessary revisions to enhance the quality of our manuscript. We answer (A) every question (Q) by the reviewers one by one as follows.

To Reviewer #1

Q1: *I encourage the authors to justify more carefully one of their stopping criteria for the searches, i.e. the number of calls to the single-step model. This seems to insert a bias against MCTS that the authors are aware of, i.e. that MCTS generally calls the single-step model more often. I think it would be better to just impose a time limit and then report search time as an evaluation criteria.*

A1: We sincerely appreciate your valuable suggestion. We conduct an experiment by imposing a time limit on running every method. A* search and MCTS are included for comparisons, and they are implemented by setting $K_{MCTS} \rightarrow \infty$ or $K_{MCTS} = 1$ in MEEA*. The results of the *success rates* and the *first solution time* are reported in Table 1, where the first solutions time means the time used to find the first feasible solution. MCTS typically takes the longest time to find the first feasible solution, due to its compulsory exploration to the unnecessary nodes. MEEA* not only solves all molecules with 100.0% success rate, but also only requires 19.93 seconds on average, significantly less than the 55.33s for MCTS and 22.99s for A*. The results again demonstrate the high efficiency of the proposed MEEA*. These experimental results have been added to Section 2.4 of the revised paper.

Table 1 The results by imposing a time limit of 10 minutes on every run.

	Success rate	First solution time
Retro*+	96.84%	43.67s
A* ($K_{MCTS} \rightarrow \infty$)	99.47%	22.99s
MCTS ($K_{MCTS} = 1$)	97.37%	55.33s
MEEA* ($K_{MCTS} = 100$)	100.0%	19.93s

It is observed that the above results are basically consistent with the ones by limiting the number of calls to the one-step retrosynthetic models. The reasons are as follows, The invocation of the one-step retrosynthetic model accounts for the majority of the runtime. When the single-step model is kept fixed, which is the scenario of the paper, capping the number of model calls is a reliable setting to evaluate the performances of retrosynthetic planning methods. Since the neural network architecture of the policy model and the single-step retrosynthetic predictor of MEEA* remain the same as that of Retro* and Retro*+, the computational resources consumed by each single-step

model invocation are basically identical. Moreover, considering the small variations in implementing various search algorithms, there exist only small differences between the runtime and the number of calls to the single-step retrosynthetic model.

Detailed comparisons of success rates under different limits on the search time are presented in Figure 1 and Table 2 below. For molecules that can be solved easily by the well-trained heuristics, A* search is the most efficient algorithm, because A* has less additional unnecessary exploration. Hence, under a small limit on the search time, A* achieves the highest success rate in Figure 1 and Table 2, while MEEA* is comparable to or slightly worse than A*. For the remaining challenging molecules, all algorithms need more time to find a feasible solution. MEEA* efficiently identified feasible solutions for all molecules with the shortest time by incorporating appropriate exploration. A* is trapped in non-optimal branches and it requires more time than MEEA* to escape. The time required by MCTS is the longest because of its compulsory too much exploration. All algorithms are guided by the same heuristics, and are capable of finding the feasible solution. Without caching the calls to the single-step model, more time is required by Retro*+ (Maziarz et al. 2023 arxiv:2310.19796). More details are provided in the Appendix I.

Figure 1 Comparison of success rates with different search time

Table 2 Success rates with different search time.

Time(s)	10	30	60	120	240	360	480	600	1200	1800
Retro*+	47.37%	68.95%	83.16%	90.00%	90.00%	93.68%	94.74%	96.84%	98.95%	99.47%
MCTS	44.74%	65.79%	74.21%	82.63%	91.58%	93.68%	97.37%	97.37%	98.95%	98.95%
A*	56.32%	85.79%	91.58%	96.32%	97.37%	98.95%	99.47%	99.47%	99.47%	99.47%
MEEA*	56.32%	81.05%	93.16%	98.42%	99.47%	100.0%	100.0%	100.0%	100.0%	100.0%

Q2: The authors mainly compare the algorithms using the number of solved targets and the length of the routes. I encourage the authors to look into other evaluation criteria as has been suggested in the literature (Genheden & Bjerrum, Dig Discov 2022; Maziarz et al. 2023 arxiv:2310.19796). The criteria that the author have chosen are insufficient evaluation criteria.

A2: We greatly appreciate the references and other evaluation criteria provided by the reviewer. Besides the number of solved targets and the length of the routes, the diversity of the generated routes is highly recommended as a criterion in the literature. For a given molecule, multiple diverse synthetic pathways are desired to be provided to human experts for further selection. In the revised paper, we include the diversity as an evaluation criterion in the experiments. Following the literature, the generated routes are clustered into different groups and the number of clusters is regarded as a metric of diversity (Genheden & Bjerrum, Dig Discov 2022). Tree edit distance (TED) is calculated to measure the similarity between the algorithm-generated synthetic pathway and the referenced expert pathway, and the leaves overlap is computed as the average ratio of the building blocks present in both the generated route and the reference route. The expert synthetic routes, which are provided for molecules in the USPTO benchmark, are used as the reference route to compute TED and the leaves overlap. During the experiment, at most 25 synthetic routes are generated to compute the number of clusters, because it becomes slow if a large number of routes are fed into the clustering algorithm by (Genheden & Bjerrum, Dig Discov 2022). The TED value and the leaves overlap are calculated based on the first generated route. According to the results in Table 3, the routes provided by MCTS have greater diversity with larger cluster numbers. Conversely, the routes generated by A* are more closely aligned with the expert route with a smaller TED and a higher proportion of overlapping leaves. MEEA* is a hybrid algorithm that combines the characteristics of A* and MCTS, and it achieves a good balance between the route quality and the diversity. These experimental results have been added to Section 2.4 of the paper.

Table 3 The quality and diversity of the predicted routes

	Number of clusters	TED to expert	Leaves overlap
Retro*+	/	11.10	0.606
A* ($K_{MCTS} \rightarrow \infty$)	4.09	10.15	0.638
MCTS ($K_{MCTS} = 1$)	4.26	11.27	0.582
MEEA* ($K_{MCTS} = 100$)	4.17	10.94	0.605

Retro*+ is not designed to generate multiple synthetic pathways, and only the best route is provided. The number of clusters for Retro*+ is not provided.

In the literature, PaRoutes (Genheden & Bjerrum, Dig Discov 2022) collected 10,000 molecules and their synthetic routes as a test set for evaluating the performances of retrosynthetic planning methods. However, it should be noted that these molecules were selected from the USPTO dataset, which has already been utilized for training by many algorithms, such as Retro* and Retro*+. What's more, the routes with a depth of more than 10 reactions have been excluded in PaRoutes. MCTS exhibits poor

performance on these discarded molecules with longer paths as discussed in our paper, and this selection process may introduce a bias in favor of MCTS search. Actually, 15.3% of the routes in the USPTO test benchmark exceed a length of 10. Therefore, the test molecule set employed in PaRoutes may not be suitable as a benchmark for evaluating retrosynthetic planning algorithms.

Q3: *Line 64, in introduction claims "demonstrated more promising results" but both "Genheden & Bjerrum, Dig Discov 2022" and "Maziarz et al. 2023 arxiv:2310.19796" gives a more nuanced view of the relative merit of MCTS and Retro* and this should be mentioned in the introduction.*

A3: Thanks for your suggestion. More detailed comparative results in previous works are added to our paper. As mentioned in Line 74-77 of the manuscript, MCTS and Retro* have demonstrated comparable search speeds and capabilities in route discovery. When comes to the route quality and diversity, MCTS outperforms Retro*.

Q4: *On page 8, the link to e-molecule stock is dead as the company does not provide the downloads from previous years.*

A4: We appreciate your reminder and have updated the hyperlink to the accessible version <http://downloads.emolecules.com/free/2023-12-01/>.

Q5: *Line 194, page 8 states "It is important to note that these molecules may not necessarily have feasible synthesis routes among the top 50 chemical reactions," but what exactly the author means by this needs to be clarified.*

A5: The USPTO benchmark requires that each molecule in the test set possesses at least one synthetic pathway, and the pathway is constructed by the reactions ranking within the top 50 predictions of the single-step expansion model. The molecules in the nine additional datasets are not subject to these restrictions, and may not have feasible synthesis routes among the top 50 chemical reactions, which renders them more representative of real-world molecule distributions. We have made revisions to the paper to enhance its clarity in Line 216-222

To Reviewer #2

Q1: *The introduction section can be improved by drawing comparisons to similar work in the field. While there are comparisons to existing work, it is the reviewer's opinion that currently the manuscript does not provide adequate references to existing methods.*

A1: Thank you very much for your suggestion. We have incorporated 18 references into the paper to give a more comprehensive summary. The details are listed below.

In Line 39-46, three papers about template-based one-step retrosynthetic prediction models have been added:

- [9] Coley, C.W., Barzilay, R., Jaakkola, T.S., Green, W.H., Jensen, K.F.: Prediction of organic reaction outcomes using machine learning. *ACS central science* 3(5), 434–443 (2017)
- [10] Dai, H., Li, C., Coley, C., Dai, B., Song, L.: Retrosynthesis prediction with conditional graph logic network. *Advances in Neural Information Processing Systems* 32 (2019)
- [11] Coley, C.W., Rogers, L., Green, W.H., Jensen, K.F.: Computer-assisted retrosynthesis based on molecular similarity. *ACS central science* 3(12), 1237–1245 (2017)

Five papers about template-free one-step model have been added:

- [13] Lin, K., Xu, Y., Pei, J., Lai, L.: Automatic retrosynthetic route planning using template-free models. *Chemical science* 11(12), 3355–3364 (2020)
- [14] Schwaller, P., Laino, T., Gaudin, T., Bolgar, P., Hunter, C.A., Bekas, C., Lee, A.A.: Molecular transformer: a model for uncertainty-calibrated chemical reaction prediction. *ACS central science* 5(9), 1572–1583 (2019)
- [15] Zheng, S., Rao, J., Zhang, Z., Xu, J., Yang, Y.: Predicting retrosynthetic reactions using self-corrected transformer neural networks. *Journal of chemical information and modeling* 60(1), 47–55 (2019)
- [16] Somnath, V.R., Bunne, C., Coley, C.W., Krause, A., Barzilay, R.: Learning graph models for template-free retrosynthesis. *arXiv preprint arXiv:2006.07038* (2020)
- [17] Tetko, I.V., Karpov, P., Van Deursen, R., Godin, G.: State-of-the-art augmented nlp transformer models for direct and single-step retrosynthesis. *Nature communications* 11(1), 5575 (2020)

Three papers about semi-template-based methods have been added. They predicted the reaction center dictating a reaction firstly via graph neural networks, and then translated the resulted intermediate synthons into reactants via translation models:

- [18] Sacha, M., Bl az, M., Byrski, P., Dabrowski-Tumanski, P., Chrominski, M., Loska, R., Wl odarczyk-Pruszynski, P., Jastrzebski, S.: Molecule edit graph attention network: modeling chemical reactions as sequences of graph edits. *Journal of Chemical Information and Modeling* 61(7), 3273–3284 (2021)
- [19] Shi, C., Xu, M., Guo, H., Zhang, M., Tang, J.: A graph to graphs framework for retrosynthesis prediction. In: *International Conference on Machine Learning*, pp. 8818–8827 (2020). PMLR
- [20] Yan, C., Ding, Q., Zhao, P., Zheng, S., Yang, J., Yu, Y., Huang, J.: Retroxpert: Decompose retrosynthesis prediction like a chemist. *Advances in Neural Information Processing Systems* 33, 11248–11258 (2020)

Retrosynthetic planning algorithms with greedy strategy or other simple exploration strategies have been considered by adding three related papers:

- [21] Schreck, J.S., Coley, C.W., Bishop, K.J.: Learning retrosynthetic planning

through simulated experience. ACS central science 5(6), 970–981 (2019)

[22] Schwaller, P., Petraglia, R., Zullo, V., Nair, V.H., Haeuselmann, R.A., Pisoni, R., Bekas, C., Iuliano, A., Laino, T.: Predicting retrosynthetic pathways using transformer-based models and a hyper-graph exploration strategy. Chemical science 11(12), 3316–3325 (2020)

[23] Tripp, A., Maziarz, K., Lewis, S., Segler, M., Hernández-Lobato, J.M.: Retro-fallback: retrosynthetic planning in an uncertain world. arXiv preprint arXiv:2310.09270 (2023)

Two papers about depth-first proof number search are also considered:

[28] Kishimoto, A., Buesser, B., Chen, B., Botea, A.: Depth-first proof-number search with heuristic edge cost and application to chemical synthesis planning. Advances in Neural Information Processing Systems 32 (2019)

[29] Franz, C., Mogk, G., Mrziglod, T., Schewior, K.: Completeness and diversity in depth-first proof-number search with applications to retrosynthesis. In: 31st International Joint Conference on Artificial Intelligence, IJCAI 2022, pp. 4747–4753 (2022). International Joint Conferences on Artificial Intelligence

Two papers about publicly available retrosynthetic planning software tools have been added:

[35] Genheden, S., Thakkar, A., Chadimova, V., Reymond, J.-L., Engkvist, O., Bjerrum, E.: Aizynthfinder: a fast, robust and flexible open-source software for retrosynthetic planning. Journal of cheminformatics 12(1), 70 (2020)

[36] Latendresse, M., Malerich, J.P., Herson, J., Krummenacker, M., Szeto, J., Vu, V.-A., Collins, N., Madrid, P.B.: Synroute: A retrosynthetic planning software. Journal of Chemical Information and Modeling 63(17), 5484–5495 (2023)

Q2: *Y-axis label in figure 2 is slightly confusing.*

A2: Thank you for pointing out this issue. The purpose is to quantify the proportions of the failed molecules that can be synthesized by exhaustive search using reactions in USPTO dataset within a given number of synthetic steps. The failed molecules are referred to those that are failed to search for a synthetic pathway by the given method. We have replaced the original distribution histogram with a plot of cumulative proportions. The X-axis represents the number of steps required for exhaustive search to synthesize the molecules, while the Y-axis depicts the cumulative proportion of the molecules that are failed to search for a synthetic pathway by the given method. For example, considering the failed molecules by Retro*, 78.95% of them can actually be synthesized within 7 steps by exhaustive search, and 89.47% can be synthesized within 9 steps. This observation suggests that A* search may delve too deeply into unproductive branches, and fail to find the solution under limited search time.

Figure 2 The cumulative proportions of lengths of synthetic routes required by the expert for molecules that are failed to be synthesized by the search algorithm

Q3: Can the authors comment on how sensitive is the performance to the MCTS simulation?

A3: We conduct an experiment on the sensitivity of the performance to the MCTS simulation times as given by the figure below. The results indicate that there is a wide range for K_{MCTS} make MEEA* perform well. MCTS simulation times K_{MCTS} is set to balance the influence of A* search and MCTS on MEEA*. When K_{MCTS} is relatively small, MEEA* approaches to MCTS. When K_{MCTS} is quite large, MEEA* closely resembles A* search. As shown in Figure 3, both extreme scenarios exhibit a reduction in performance. Appropriate K_{MCTS} achieves a better balance between exploration and exploitation, resulting in improved performance. Discussions are included in Line 448-449. More details are provided in the Appendix H.

Figure 3 Success rate of MEEA* with different K_{MCTS}

Q4: *The main text does not provide any detailed descriptions of the used neural networks, although these are main components of the introduced workflow. Adding implementation details to the manuscript will be highly useful.*

A4: Thank you for pointing this out, and we add the implementation details of the neural networks in Appendix C. There are two neural networks employed in this paper: a policy network in the single-step retrosynthetic model **B** and a value network for state evaluation. The architecture of the policy network is identical to Retro* and Retro*+. The input molecule is represented by its Morgan fingerprint, which is a vector of 2048 dimension. The output is a probability distribution over all available 381302 chemical templates. The policy network is composed of:

- A fully connected layer [2048, 512].
- A batch normalization layer.
- A dropout layer with a dropout rate of 0.3.
- A fully connected layer [512, 381302].

When estimating synthetic cost independently in Equation 5, the architecture of the value network is also identical to Retro* and Retro*+, which is composed of:

- A fully connected layer [2048, 128].
- A ReLU activation layer.
- A dropout layer with a dropout rate of 0.1.
- A fully connected layer [128, 1].

When estimating synthetic cost jointly in Equation 6, the input of the value network consists of multiple molecules. A summation layer is utilized to perform elementwise summation of the extracted features of all molecules, producing a global feature representation. The architecture is composed of:

- A fully connected layer [2048, 128].
- A ReLU activation layer.
- A dropout layer with a dropout rate of 0.1.
- A summation layer to produce global representation $N \times 128 \rightarrow 1 \times 128$.
- A fully connected layer [128, 1].

Q5: *What is meant by “The performance of the expert is based on the pathways provided by the USPTO benchmark, which is the best synthetic route using all reactions in USPTO.”? If the performance is based on the USPTO benchmark, then how is it computed?*

A5: In Table 1, the performance of the expert is presented for comparison. The synthetic pathways provided by the USPTO test benchmark are considered as the synthetic pathways of the expert. For molecules appeared in USPTO benchmark, exhaustive search is employed to collect all possible pathways for synthesizing the target molecule utilizing chemical reactions from the USPTO dataset, and the shortest route is selected as the synthetic route of the expert. After obtaining the expert's synthetic pathway, the length and cost of this pathway can be calculated. Length of a synthetic pathway is the number of reactions. Reaction cost is defined in Line

395-398, and cost for a synthetic pathway is the summation of costs of all reactions in this pathway.

Q6: *It is stated that “However, the USPTO training set provided by Retro* [13] does not include synthetic pathways for molecules, so we will generate these routes using retrosynthetic planning algorithms, serving as the training set for the following task.” Did the training data come from other retrosynthetic planning algorithms? Couldn't the pathways provided by the USPTO benchmark be used in this setting? This needs further clarification.*

A6: The USPTO training set provided by Retro* does not include synthetic pathways for molecules, so we are required to generate these routes by ourselves. One choice is to employ exhaustive search to identify the shortest synthetic pathway of the given molecule using the chemical reactions available in the USPTO dataset, the same as Retro*. This strategy is effective to train a model with a relative high performance from scratch, but limits the performance of the model, making it challenging to surpass the capabilities represented by the training set, as discussed in AlphaGo and AlphaGo Zero. The other choice is to update the heuristic model with reinforcement learning, collecting synthetic routes with MEEA* to further update its heuristic guidance iteratively. Retro*+ has illustrated the effectiveness of this self-improved learning process. Therefore, MEEA* search is utilized to identify the synthetic pathways for molecules in the USPTO training set. The expansion policy and cost estimator from Retro*+ are employed as the initial heuristic functions to assist MEEA* search to identify synthetic pathways, which will be used to update the heuristic functions of MEEA*.

Q7: *A general critique of the manuscript is that the sentences tend to be too long and repetitive. Reframing such long sentences will improve readability.*

A7: We sincerely appreciate your insightful feedback. we have diligently revised the lengthy sentences in the manuscript to enhance readability. Here are a few examples of the changes we made.

1. Original: Despite the widespread utilization of the USPTO benchmark for evaluating retrosynthetic planning algorithms, the difficulty of this benchmark has been intentionally reduced by including only molecules synthesizable with reactions in the USPTO database, where each reaction must rank within the top 50 predictions of the single-step expansion model of Retro*
Revised: USPTO benchmark is widely utilized to evaluate retrosynthetic planning algorithms. However, molecules in the test benchmark needs to satisfy two conditions. Synthetic route with reactions in the USPTO database is available, and each reaction in the synthetic route must rank within the top 50 predictions of the single-step expansion model of Retro*. These two screening criteria have intentionally reduced the difficulty of this benchmark.
2. Original: By incorporating exploratory behavior into A*, MEEA* can explore

alternative paths and avoid getting trapped in local optimal branches, while still utilizing A*'s efficient node expansion criteria to avoid the need for exhaustive exploration of all branches.

Revised: Compared to A*, MEEA* escapes local optimal branches by incorporating exploratory behavior. In contrast to MCTS, MEEA* prevents exhaustive exploration of all branches by utilizing A*'s efficient node expansion criteria.

3. Original: The single-step retrosynthetic model \mathcal{B} is to select the k most promising chemical reactions as legal actions according to the expansion policy $\pi(a|s)$, while disregarding others, i.e., $\mathcal{B}(m) = \{a_i, \pi(a_i|m)\}_{i=1}^k$, where $\pi(a_i|m)$ is the associated probability of taking action a_i .

Revised: The single-step retrosynthetic model \mathcal{B} selects the k most promising chemical reactions as legal actions i.e., $\mathcal{B}(m) = \{a_i, \pi(a_i|m)\}_{i=1}^k$, where $\pi(a_i|m)$ is the associated probability of taking action a_i for molecule m . The remaining reactions are discarded to narrow down the width of the search tree.

Q8: *A typo was found in “It is capable of talking all molecules within the current state jointly as input and computing the state cost in a direct and more accurate manner.”*

A8: Thank you very much for pointing out this error, and we have rectified it in the original manuscript.

REVIEWERS' COMMENTS:

Reviewer #1 (Remarks to the Author):

I believe the authors have incorporated well the remarks by the reviewers and therefore I believe the manuscript can be published as-is.

Reviewer #2 (Remarks to the Author):

Thank you for addressing the previous comments. Reviewer is satisfied with the current status of the manuscript.

Dear Editor and Reviewers,

Thank you for your insightful comments and suggestions regarding our manuscript. We have made revisions to the figures and tables in the paper and standardized the formatting of the supplementary materials. Your expertise and thorough evaluation have significantly improved our research work. We remain committed to continuously improving our work based on your insightful comments.

With sincere gratitude,
Dengwei Zhao

Reviewer #1 (Remarks to the Author):

I believe the authors have incorporated well the remarks by the reviewers and therefore I believe the manuscript can be published as-is.

Reviewer #2 (Remarks to the Author):

Thank you for addressing the previous comments. Reviewer is satisfied with the current status of the manuscript.